# Chaperone-Like Activity of HSPB5: The Effects of Quaternary Structure Dynamics and Crowding

**DOI:** 10.3390/ijms21144940

**Published:** 2020-07-13

**Authors:** Natalia A. Chebotareva, Svetlana G. Roman, Vera A. Borzova, Tatiana B. Eronina, Valeriya V. Mikhaylova, Boris I. Kurganov

**Affiliations:** Bach Institute of Biochemistry, Federal Research Centre “Fundamentals of Biotechnology” of the Russian Academy of Sciences, Leninsky pr. 33, 119071 Moscow, Russia; svetabaj@gmail.com (S.G.R.); squsiebox@gmail.com (V.A.B.); eronina@inbi.ras.ru (T.B.E.); mikhaylova.inbi@inbox.ru (V.V.M.); boris@kurganov.com (B.I.K.)

**Keywords:** HSPB5, chaperone-like activity, oligomeric states, mixed crowding

## Abstract

Small heat-shock proteins (sHSPs) are ATP-independent molecular chaperones that interact with partially unfolded proteins, preventing their aberrant aggregation, thereby exhibiting a chaperone-like activity. Dynamics of the quaternary structure plays an important role in the chaperone-like activity of sHSPs. However, relationship between the dynamic structure of sHSPs and their chaperone-like activity remains insufficiently characterized. Many factors (temperature, ions, a target protein, crowding etc.) affect the structure and activity of sHSPs. The least studied is an effect of crowding on sHSPs activity. In this work the chaperone-like activity of HSPB5 was quantitatively characterized by dynamic light scattering using two test systems, namely test systems based on heat-induced aggregation of muscle glycogen phosphorylase *b* (Ph*b*) at 48 °C and dithiothreitol-induced aggregation of α-lactalbumin at 37 °C. Analytical ultracentrifugation was used to control the oligomeric state of HSPB5 and target proteins. The possible anti-aggregation functioning of suboligomeric forms of HSPB5 is discussed. The effect of crowding on HSPB5 anti-aggregation activity was characterized using Ph*b* as a target protein. The duration of the nucleation stage was shown to decrease with simultaneous increase in the relative rate of aggregation of Ph*b* in the presence of HSPB5 under crowded conditions. Crowding may subtly modulate sHSPs activity.

## 1. Introduction

αB-Crystallin belongs to a superfamily of small heat shock proteins (sHSPs), which are ubiquitously expressed and play an important role in maintaining cellular proteostasis [1]. sHSPs bind non-native and misfolded proteins, keeping them from further aggregation and protecting the cell from toxic aggregates [1,2,3,4,5]. In addition to the exhibition of the anti-aggregation (chaperone-like) activity, these proteins are involved in many important processes in the cell, such as apoptosis, stabilization of cytoskeleton [6], regulation of muscle contraction, regulation of redox state [7], signal transduction, etc. [4,8,9,10]. Given these significant biological roles, the dysregulation of sHSPs is associated with cancer [11], cataract formation [12,13,14], and neurodegenerative diseases [5,10]. It is known that mutations in sHSPs have been directly linked to different myopathies including Charcot-Marie-Tooth disease and neuropathy [10,14,15]. Therefore, the processes of regulation of the functioning of sHSPs are very important.

Among the ten human sHSPs, αB-crystallin (HSPB5) is one of the principle members. HSPB5 is widespread in all tissues, but its concentration in the eye lens is especially high (400 mg/mL), where it interacts with αA-crystallin (HSPB4) and forms a native hetero-oligomeric complex, α-crystallin [16]. These proteins ensure the transparency of the eye lens [17], preventing the aggregation of other crystallins and thereby protecting the lens from the development of cataracts [12]. All sHSPs, such as HSPB5, in their structure have a central α-crystalline domain (ACD) with an ordered structure, which is flanked by two variable terminal regions, C- and N- terminal domains, with a partially disordered structure, intrinsically disordered regions (IDRs) [18]. The ACD domain plays an important role in the formation of a dimer, which is considered as the building block that assembles via terminal interactions into a polydisperse ensemble under physiological conditions [9,19,20]. N-terminal regions are involved in the formation of large oligomers [18,21,22]. Carver and coworkers proposed, that the mobile C- and N-terminal regions of HSPB5 are involved in regulation of interaction with substrates; they protect ACD domain from amyloid fibril formation [20] and the flexibility of these regions, especially C-terminal regions, provides solubility for sHSPs [18,20]. The importance of the IDRs regions in sHSPs for formation of dynamic macromolecular assemblies and regulation of protein functionality has been discussed in the work [10].

HSPB5 tends to form large polydisperse assemblies ranging from 10-mer to 40-mer and higher, having very dynamic structures [19,23,24,25]. All oligomeric forms possess chaperone-like activity and easily exchange their subunits [20,26,27]. It is well accepted that a flexible dynamic quaternary structure is necessary for sHSPs activity [1,4,10,28]. Changes in the cellular environment, such as temperature [9,10,27,29], pH [30], the presence of ions [31,32], post-translational modification (phosphorylation) [33,34,35], redox environment [7,30] and crowding [19,25,32,36,37,38] also regulate chaperone activity by affecting the structural and oligomerization dynamics of sHSPs [4,5]. It has been reported that sHSP dynamics is a very complex process that includes five levels of regulation: (1) flexible domains flanking the ACD, (2) polydisperse self-oligomerization, (3) hetero- oligomerization with other sHSPs, (4) subunit exchange, and (5) regulation by the cellular environment [5,39]. As such, multiple oligomeric forms are likely relevant to function, and regulation thereof. The importance of quaternary dynamics for realization of chaperone activity of sHsps was discussed in [5,9,10,19,28,29]. However, the relationship between the dynamic variable structure and functions of sHSPs is far from being clear. The least studied is the effect of crowding on this relationship.

The cellular interior contains large concentrations of macromolecules, reaching up to 400 g/L, including proteins, nucleic acids, lipids, glycans, and solvated ions [40,41]. This means that about 40% of cell volume may be occupied by macromolecules and become physically unavailable for other molecules. A nonspecific influence of such unavailable volume on the specific biochemical reactions [42,43] was termed macromolecular crowding or excluded volume effect [44]. Theory predicts that crowding affects both the kinetics and thermodynamics of interactions between macromolecules, including protein aggregation [38,42,43,45,46,47,48].

It is usual to mimic crowded milieu in vitro by adding high concentration of suitable inert polymer or protein, so-called crowding agent, such as polyethylene glycols (PEG) of different molecular mass, polysaccharides (Ficoll, dextran), polyvinylpyrrolidone (PVP) or proteins (bovine serum albumin, lysozyme) [42]. For a long time, researchers simulated crowding in vitro by adding a single crowding agent at concentrations >100 mg/mL. Considering that the molecules of crowders are completely neutral and the effect of crowding on biochemical reactions is manifested due to steric repulsion of macromolecules, that is, the excluded volume effect (EVE). Currently, it is believed that the influence of crowding can be considered to be a mixture of entropic (excluded volume) and enthalpy based (soft interactions) effects [47,49,50,51,52]. “Soft” interactions include electrostatic, hydrophobic, and van der Waals interactions between the crowding agent and studied protein [53,54]. These interactions can be repulsive or attractive. Therefore, such soft interactions (enthalpy factor) can counteract EVE [50,55,56,57,58].

Given that in a cell crowding is created by the presence of different molecules that differ in size, shape and charge, some groups of researchers came to the conclusion that it is better to imitate physiological conditions in vitro with a mixture of several crowders (mixed crowding) [47,49,50,51,52,59]. It has been shown that two crowders can exhibit a synergistic effect, significantly enhancing the effect of each other, even when using relatively small concentrations (10–20 mg/mL) [49]. It has also been shown that small crowders create more total excluded volume in the vicinity of big crowder than in the bulk [49,58]. Sharp pointed out that when the steric effects of macromolecular crowders and small molecules like water and ions are treated on an equal footing, the effect of the macromolecules are less effective at crowding than water and ions [60]. Shah and coworkers developed a molecular thermodynamic formalism to examine the effects of size-polydispersity of crowders on aggregation reaction equilibrium. They showed that in the case of polydisperse crowders, the crowders with the largest size difference dominate the overall changes in the yield of the reaction [50].

Thus, crowding adds another level of complexity to the relationship between the activity and structural dynamics of HSPB5. Since it has a strong effect on protein–protein interactions, it should affect the conformation and self-association of the chaperone, the interaction of the chaperone with the target protein, and the aggregation of the target protein. Previously, we have shown by the analytical ultracentrifugation (AUC) method that crowding strongly affects the oligomeric state of HSPB5, HSPB6, HSPB1 and α-crystallin [25,32,36,61,62,63,64]. Thus, one might assume that crowding affects the capability of sHSPs to prevent aggregation of target proteins.

The goal of this work was to quantitatively assess the effect of crowding, including mixed crowding, on the chaperone-like activity of HSPB5. As the process of protein aggregation includes the stage of protein unfolding followed by the aggregation of unfolded protein molecules, in a general case, sHSPs can affect the unfolding stages as well as the aggregation stage. Therefore, it is very important to question what stage of the overall aggregation process (unfolding or aggregation) is rate-limiting in the selected test system.

In the present work, two test systems were selected to study the chaperone-like activity of HSPB5, one of which is based on the thermal aggregation of glycogen phosphorylase *b* (Ph*b*) at 48 °C, and the other is based on dithiothreitol-induced aggregation of α-lactalbumin at 37 °C; these test systems were described earlier [65,66,67,68]. Ph*b* exists as a dimer consisting of two identical subunits with molecular mass of 97.4 kDa each [69]. α-Lactalbumin (αLa) is a small Ca^2+^-binding protein containing four disulfide bridges with molar mass of 14.2 kDa. Under stress conditions, the Ca^2+^-depleted form of α-lactalbumin attains a classical molten globule state that aggregates amorphously [70,71,72,73]. The molten globule conformation of α-lactalbumin is thought to be a target for interacting with sHSPs [66,68,70,71,72,73,74,75,76]. The rate-limiting stages were established for the aggregation process for both proteins and the effect of HSPB5 on aggregation of these proteins was quantitatively evaluated.

We compared the effect of crowding by both individual crowders and pairs of crowders on the anti-aggregation activity of HSPB5 using Ph*b* as a target protein. Four pairs of crowders demonstrated a synergistic effect on the activity of HSPB5. This study has provided insights into the mechanism of chaperone function under crowded conditions.

## 2. Results

### 2.1. Protein Aggregation. Theoretical Analysis of the Initial Parts of the Kinetic Curves

One of the distinctive features of the kinetics of protein aggregation is the existence of the lag phase on the kinetic curves demonstrating enhancing the aggregation rate in the course of accumulation of protein aggregates. The appearance of the lag phase may be an indication of fulfilment of the mechanism termed “nucleation-dependent aggregation”. According to this mechanism, the initial stage of the process of protein aggregation is the stage of cooperative association of unfolded protein molecules, which results in the formation of nuclei [77,78,79,80]. The initial steps of the sticking of unfolded protein molecules are thermodynamically unfavorable. However, owing to contacts of each protein molecule with several neighbors in larger associates the change in the free energy becomes negative in the course of enlargement of the associate.

Theoretical studies predict that accumulation of the aggregated protein during nucleation stage is proportional to time squared [78]. Chen et al. [81] obtained experimental confirmation of such a regularity. When the kinetics of the protein aggregation is registered by measuring the increment of the light scattering intensity (*I*), the following quadratic equation can be used for a description of the part of the kinetic curve corresponding to the nucleation stage [82,83,84] (Figure 1A):(1)$$I-I_{0}=K_{\mathrm{agg}}{\left(t-t_{0}\right)}^{2}$$
where *I*_0_ is an initial value of the light scattering intensity at *t* = 0 and *t*_0_ is the moment of time at which the initial increment of the light scattering intensity is registered. *K*_agg_ is a parameter characterizing the acceleration of the aggregation process during the nucleation stage.

The nuclei appeared at *t* = *t*_0_ were called start aggregates [85]. To determine the hydrodynamic radius of start aggregates (*R*_h,0_), a plot of *I* versus *R*_h_ can be constructed [86,87] (Figure 1B).

The determination of the parameter *K*_agg_ value at different initial protein concentrations, [P]_0_, allows calculating the order of aggregation with respect to the protein (*b*) [84] (Figure 1C):(2)$$K_{\mathrm{agg}}=const{\left[P\right]}_{0}^{b}$$

To quantitate the anti-aggregation activity of a chaperone that form a tight complex with their targets, we used a parameter characterizing the initial adsorption capacity of the chaperone (*AC*_0_) [83,84,88,89]. To determine *AC*_0_, we need to analyze the dependence of the (*K*_agg_/*K*_agg,0_)^1/*b*^ ratio (*K*_agg,0_ is the value of parameter *K*_agg_ in the absence of the chaperone) on the ratio of molar concentrations of the chaperone and a target protein (*x* = [chaperone]/[target protein]). The stoichiometry of the chaperone–target protein complex (*S*_0_) can be determined as a length on the abscissa axis cut off by the linear part of the dependence of (*K*_agg_/*K*_agg,0_)^1/*b*^ on *x* (Figure 1D):(3)$${\left(K_{\mathrm{agg}}/K_{\mathrm{agg},0}\right)}^{1/b}=1-x/S_{0}=1-AC_{0}x$$

The *AC*_0_ value can be calculated as a reciprocal to the *S*_0_:(4)$$AC_{0}=1/S_{0}$$

In this work, we selected the parameter *K*_agg_ with the aim (1) to characterize the effect of the chaperone on the aggregation kinetics using two model substrates, namely Ph*b* (for testing thermal aggregation) and αLa (for testing DTT-induced aggregation) and (2) to quantitatively assess the effect of crowding on the chaperone-like activity of HSPB5.

### 2.2. The Effect of HSPB5 on Thermal Aggregation of Phb

Figure 2A shows the kinetic curves of thermal aggregation of Ph*b* at 48 °C (0.03 M Hepes buffer, 0.1 M NaCl, 0.2 mM EDTA, pH 6.8) registered by measuring the increment of the light scattering intensity (*I*) at several protein concentrations. The initial parts of the kinetic curves can be approximated by the quadratic Equation (1). As it can be seen from Figure 2B, there is no change in the *t*_0_ value at varying Ph*b* concentration. The average value of *t*_0_ measured at Ph*b* concentrations ranging from 0.1 to 0.9 mg/mL was found to be 136 ± 4 s. The dependence of *K*_agg_ on Ph*b* concentration is linear (Figure 2C), i.e., the coefficient *b* in Equation (2) is equal to unity. This means that the unfolding of the Ph*b* molecule proceeds with a substantially lower rate than the subsequent stages of aggregation of unfolded protein molecules. Thus, we may use the linear calibration curve {*K*_agg_; [Ph*b*]_0_} for estimating the effect of different agents (namely HSPB5 in the present work) on the initial stages of thermal aggregation of the protein.

When aggregation of Ph*b* at 48 °C was monitored in the presence of HSPB5, the retardation of the aggregation process was observed. Figure 3A demonstrates the initial parts of the dependences of the light scattering intensity on time obtained at several concentrations of Ph*b* in the presence of HSPB5 at the concentration of 0.025 mg/mL. Parameters *t*_0_ and *K*_agg_ for these dependences were estimated using Equation (1). 

The *t*_0_ value is not markedly changed in the range of Ph*b* concentrations from 0.15 to 0.6 mg/mL, being approximately 280 s, and slightly decreases to 218 s at [Ph*b*] = 0.75 mg/mL (Figure 3B). Just like in the case of Ph*b* aggregation in the absence of the chaperone, the dependence of the parameter *K*_agg_ on [Ph*b*] appeared to be linear in the presence of HSPB5 (Figure 3C) yet the slope of the line decreased by a factor of 3.5.

Figure 4 shows that the protective action of HSPB5 on the thermal aggregation of Ph*b* increases with increasing the sHSP concentration. As we can see in Figure 4A, the growth of the light scattering intensity becomes more flattened as the concentration of HSPB5 rises from 0.005 to 0.2 mg/mL. Analysis of the *I*(*t*) dependences using Equation (1) showed that the *t*_0_ value increases with increasing HSPB5 concentration and becomes three times as much when HSPB5 concentration increases from 0 to 0.2 mg/mL (Figure 4B). As for parameter *K*_agg_, its value diminishes in a non-linear fashion when the concentration of HSPB5 increases (Figure 4C).

To characterize the anti-aggregation activity of HSPB5, the (*K*_agg_/*K*_agg,0_)^1/*b*^ ≡ *K*_agg_/*K*_agg,0_ (at *b* = 1) versus *x* = [HSPB5]/[Ph*b*] plot was constructed (Figure 5, *K*_agg,0_ is the *K*_agg_ value measured in the absence of HSPB5). From the initial linear part of this dependence the initial adsorption capacity of HSPB5 (*AC*_0_) with respect to Ph*b* may be calculated as the reciprocal value of the length cut off by the straight line on the abscissa axis: *AC*_0_ = 1/*S*_0_ = (3.2 ± 0.2) monomers of Ph*b* per one subunit of HSPB5. Thus, if we assume that HSPB5 forms tight complexes with Ph*b* as it is heated at 48 °C, there might be 3.2 Ph*b* monomers per one subunit of HSPB5 in such complexes.

As one can see in Figure 5, the full dependence of *K*_agg_/*K*_agg,0_ on [HSPB5]/[Ph*b*] is not linear. The complicated shape of the plot is probably due to the dynamic structure of HSPB5 and the initial part of this dependence corresponds to the complexes of the dissociated forms of the chaperone with the target protein. The non-linear part of this dependence corresponds to the formation of the HSPB5–target protein complexes where the adsorption capacity of HSPB5 in respect to the target protein becomes decreased.

Analysis of the DLS data allows for a conclusion that the observed increase in the light scattering intensity with time (Figure 4A) is caused by the growth of Ph*b* aggregates in size. Figure 6A shows the time dependences of the hydrodynamic radius (*R*_h_) of protein aggregates formed in the course of thermal aggregation of Ph*b* in the absence and in the presence of HSPB5. Rising the concentration of HSPB5 from 0.005 to 0.2 mg/mL results in evident decrease in the size of registered aggregates.

It has been shown earlier that there is a linear relationship between the light scattering intensity (*I–I*_0_) and the *R*_h_ value for the initial parts of the kinetic curves observed for aggregation of Ph*b* at 48 °C [65]. In the presence of HSPB5 the dependences of (*I–I*_0_) on *R*_h_ still be linear at the beginning of the aggregation process (Figure 6B). The length cut off on the abscissa axis by the linear dependence of (*I–I*_0_) on *R*_h_ corresponds to the hydrodynamic radius of the start aggregates (*R*_h,0_). The inset in Figure 6B represents the *R*_h,0_ values obtained for aggregation of Ph*b* in the presence of HSPB5 at various concentrations of the latter. The *R*_h,0_ value was found to be 30 ± 6 nm when Ph*b* aggregation was studied in the absence of HSPB5. The addition of HSPB5 at the concentration of 0.05 mg/mL results in more than 2.5-times size reduction of the start aggregates, and the *R*_h,0_ value does not change with further increase in HSPB5 concentration up to 0.2 mg/mL.

### 2.3. The Effect of HSPB5 on DTT-Induced Aggregation of α-Lactalbumin (αLa)

To characterize the duration of the lag phase (*t*_0_) and the initial rate of the stage of aggregates growth, Equation (1) also can be used in the case of αLa aggregation (0.1 M Na-phosphate buffer, pH 6.8) at 37 °C in the presence of 20 mM DTT [66]. The dependence of the parameter *K*_agg_ on αLa concentration is linear in coordinates {log([αLa]); log(*K*_agg_)} (Figure 7A). The slope of the linear fitting is equal to parameter *b* in Equation (2), (*b* = 4.9 ± 0.5). Thus, the stages of aggregation of αLa become the rate limiting in the whole aggregation process, so that this test system allow testing the direct effect of various agents on the stage of aggregation of unfolded protein molecules [84].

Since it is known that the activity of a chaperone can be influenced/regulated by different target proteins, we compared the effect of HSPB5 on the kinetics of thermal aggregation of Ph*b* with its effect on the kinetics of DTT-induced aggregation of αLa. The comparison of test systems with different kinetic regimes and/or aggregation mechanisms can give an additional insight in the anti-aggregation functioning of chaperones, in this case in the functioning and related structural properties of sHSPs.

Figure 7B shows the dependences of the light scattering intensity (*I*–*I*_0_) on time for DTT-induced aggregation of αLa (1.0 mg/mL) in the absence (curve 1) and presence of different concentrations of HSPB5 (curves 2–5). The suppression of the increment of the light scattering intensity in time with increasing HSPB5 concentration was observed and can be characterized by parameters *K*_agg_ and *t*_0_. As in the case of Ph*b*, the dependence of the relative parameter (*K*_agg_/*K*_agg,0_)^1/*b*^ on the ratio of the molar concentrations of HSPB5 and αLa was plotted (Figure 7C). The initial stoichiometry of the HSPB5–αLa complexes (*S*_0_) determined using Equation (3) was found to be equal to 0.011 ± 0.001. Thus, we may evaluate the adsorption capacity of HSPB5 with respect to αLa at the initial stages of aggregation: *AC*_0_ = 1/*S*_0_ = (91 ± 8) αLa molecules per one HSPB5 subunit.

### 2.4. Sedimentation Velocity (SV) Analysis of HSPB5 and α-Lactalbumin at 37 °C

SV analysis made it possible to verify the interaction of αLa with HSPB5 at low concentrations of the latter. Figure 8 shows the differential sedimentation coefficient distribution, *c*(*s*), for αLa denatured by the addition of 20 mM DTT at 37 °C. The *c*(*s*) distribution exhibits a broad peak with a sedimentation coefficient (*s*_20,w_) of 2.8 S and a minor peak (~3%) with *s*_20,w_ of 6.1 S, which indicates the presence of a mixture of denatured monomers, dimers and small aggregates of αLa. 

The data obtained by asymmetric field flow (Appendix A) also show that after 3 min incubation at 37 °C in the presence of 20 mM DTT, denatured αLa exists during 30 min as a mixture of monomers, dimers and small aggregates.

In the case when HSPB5 at concentration of 0.00045 mg/mL was added, the *c*(*s*) distribution shows the main narrow peak with *s*_20,w_ = 1.9 S. Unfortunately, it is impossible to verify the oligomeric state of HSPB5 at such a low chaperone concentration by the SV method using an absorption optical system, but earlier we showed that the dissociation of HSPB5 occurs when the chaperone concentration decreases [32]. It can be assumed that the chaperone exists in a dimeric or monomeric form at such a low concentration. The *c*(*s*) distribution of αLa is shifted toward smaller values on the addition of HSPB5, indicating the formation of hetero-complexes. With an increase in the concentration of HSPB5, the position of the peak in the *c*(*s*) distribution changes and the peak splits into 2 peaks with the sedimentation coefficients of 1.6 and 2.0 S at the HSPB5 concentration of 0.03 mg/mL (or 2.1 µM). The data obtained indicate that the αLa dimer may dissociate into monomers in the presence of the chaperone during denaturation. The interaction of HSPB5 with αLa results in the formation of small hetero-oligomers that differ in the molar mass and shape. The molar masses estimated by SEDFIT are given in Table 1. The molecular mass varies from 25 to 37–47 kDa with an increase in the concentration of the chaperone. The value of the molar mass equal to 26 kDa can correspond to dimers of the target protein, taking into account that the molecular mass of αLa monomer is 14.2 kDa, and values of the molar mass of 37–47 kDa can correspond to complexes between αLa monomer and HSPB5 monomer (with a subunit mass of 20 kDa) or complexes between a dimer of the target protein and a monomer of the chaperone (47 kDa). However, the existence of high-order oligomeric HSPB5–αLa complexes cannot be ruled out. Such complexes may be present in DLS measurements. The typical particle size time-dependences for aggregates of αLa (1.0 mg/mL) in the absence or presence of HSPB5 are shown in Appendix A.

### 2.5. Sedimentation Velocity Analysis of HSPB5 and Phb at 48 °C

Figure 9 shows the sedimentation behavior of Ph*b* in the presence of low concentrations of HSPB5 at 48 °C. The *c*(*s*) distribution for Ph*b* (0.5 mg/mL) exhibits one main peak and several small peaks, which may correspond to a partially unfolded dimer (*s*_20,w_ = 7.7 S), the denatured monomer (*s*_20,w_ = 6.6 S) and a small amount of the native dimer (*s*_20,w_ = 8.6 S). In addition, during the acceleration of the rotor, about 17% of the protein aggregated and precipitated. In the presence of even small concentrations of HSPB5, the position of the main peak (7.7 ± 0.1 S) is shifted toward the higher values (Figure 9, *s*_20,w_ = 8.0 ± 0.1 S and 8.2 ± 0.2 S in the presence of 0.025 and 0.075 mg/mL HSPB5, respectively), and the positions of minor peaks change, that indicates the interaction of the chaperone with the target protein. As for the main peak, the sedimentation coefficient value increases, that means an increase in the molecular mass of the complex, on the one hand, and a change in the conformation of the target protein, on the other. The molecular mass estimate for the main peaks of Ph*b* in the absence and presence of HSPB5 at concentrations of 0.025 and 0.075 mg/mL (Figure 9) using Svedberg equation (see Section 4.3) gives the values: 205 ± 10, 234 ± 10 and 230 ± 10 kDa, respectively. For calculations, we used sedimentation coefficients determined by AUC and diffusion coefficients (*D*), determined by DLS at 48 °C. The coefficients *D* for Ph*b* and possible complexes of Ph*b* with HSPB5 were estimated to be (6.5 ± 0.3) × 10^−7^, (5.9 ± 0.5) × 10^−7^, (6.2 ± 0.3) × 10^−7^ cm^2^/s. The *R*_h_ values obtained by DLS method were equal to 6.2, 6.9 and 6.4 nm, respectively. The data of SV analysis and data obtained by DLS method indicate the possibility of the formation of complexes between the dimeric form of the Ph*b* and the monomeric or dimeric form of HSPB5.

### 2.6. Sedimentation Velocity Analysis of HSPB5 and Phb at 48 °C Under Crowded Conditions

A comparison of the *c*(*s*) distributions for Ph*b* and the mixtures of Ph*b* and HSPB5 under crowded conditions arising from the presence of PEG_20kDa_ (25 mg/mL) revealed a shift in the position of the major peak from *s*_20,w_ = 10.3 S for Ph*b* to *s*_20,w_ = 9.8 S for the mixture (Figure 10). 

This shift, combined with the increasing area under the major peak on the *c*(*s*) distributions for the mixtures of HSPB5 and Ph*b* (Figure 10, red and blue curves), indicates the interaction between the chaperone and the target protein under crowded conditions. Crowding can both stabilize the target protein at the denaturation stage of the aggregation process and accelerate the process at the aggregation stage. The amount of the protein aggregated and precipitated during rotor acceleration increased from 17%, in the case of sedimentation of Ph*b* in the dilute solution (Hepes buffer), up to 37% for Ph*b* sedimentation in the presence of PEG_20kDa_.

The addition of HSPB5 protects Ph*b* from precipitation. The fraction of the precipitated protein decreases by almost twofold (up to 19%) in the presence of the chaperone (data not shown). This also indicates the interaction between the chaperone and the target protein under crowded conditions. Although we cannot obtain the sedimentation behavior of HSPB5 itself at the concentrations used, according to our earlier data HSPB5 appears to dissociate at elevated temperature in a concentration dependent manner [25,32]. Thus, HSPB5 may be in a dissociated form (dimeric or monomeric) at such low concentrations in a buffer solution. However, under crowded conditions, the chaperone assembles into large-sized oligomers at 48 °C [25].

The interaction of HSPB5 with Ph*b* under the mixed crowding and the heat stress conditions was studied by the sedimentation velocity analysis. The data of SV analysis are presented in Figure 11. In the case of HSPB5 the *c*(*s*) distribution contains the main narrow peak with *s*_20,w_ = 23 S, which is consistent with our earlier data [25]. This value, 23 S, corresponds to a large oligomer assembly with the molar mass of 621 kDa [25]. The *c*(*s*) distribution for partially denatured Ph*b* reveals several peaks with sedimentation coefficients of 7, 9, 12 S and several minor peaks corresponding to small aggregates. A comparison of individual distributions for the chaperone and the target protein with that for the mixture of HSPB5 and Ph*b* reveals that the peak at 23 S disappears, the peak at 7 S also disappears, the area under the peak at 9 S decreases significantly in the *c*(*s*) distribution for the mixture. At the same time new peaks appear for the mixture with the values of *s*_20,w_ less or greater than 23 S, namely, peaks at 11, 14, 16 S and 25, 31 S, respectively. These data can be explained by the dissociation of the chaperone upon encountering a target protein and the formation of complexes of different sizes and masses between the suboligomeric forms of HSPB5 and Ph*b*.

### 2.7. Effect of Crowding on the Chaperone-Like Activity of HSPB5

The effects of four macromolecular crowders (PEG_20kDa_, PVP_10kDa_, PVP_25kDa_, Ficoll_70kDa_) and five mixed pairs of crowders (PEG_20kDa_ + Ficoll_70kDa_, PVP_10kDa_ + Ficoll_70kDa_, PVP_25kDa_ + Ficoll_70kDa_, PVP_10kDa_ + PEG_20kDa_, PVP_25kDa_ + PEG_20kDa_) on the kinetics of heat-induced aggregation of Ph*b* were evaluated by measuring the increment of the light scattering intensity with time. We selected the parameter *K*_agg_ for quantitative assessing the effect of crowding on the chaperone-like activity of HSPB5. Table 2 shows that crowding has a significant effect on the kinetic parameters of the protein aggregation and on the chaperone-like activity of HSPB5. All the crowders and their mixtures shorten the lag time *t*_0_ and increase the *K*_agg_ value and the *K*_agg_/$K_{\mathrm{agg}}^{0}$ ratio, where $K_{\mathrm{agg}}^{0}$ is the *K*_agg_ value measured in the presence of HSPB5 and in the absence of any crowders. The greatest decrease in the lag period (1.6 times) and an increase in the relative rate of aggregation of Ph*b* (the value of the parameter *K*_agg_/$K_{\mathrm{agg}}^{0}$ increases by 55 times) occurs in the presence of a mixture of two polymers (PVP_25kDa_ + PEG_20kDa_). That is, crowding reduces the chaperone-like activity of HSPB5.

When assessing the effect of mixed crowding, synergism or antagonism in the action of crowders in a pair can be evaluated. To characterize the combined action of crowding agents, parameter *j* can be proposed:(5)$$j=\frac{\left(K_{\mathrm{agg}}^{1,2}/K_{\mathrm{agg}}^{0}\right)-1}{\left[\left(K_{\mathrm{agg}}^{1}/K_{\mathrm{agg}}^{0}\right)-1]+[\left(K_{\mathrm{agg}}^{2}/K_{\mathrm{agg}}^{0}\right)-1\right]}$$
where $K_{\mathrm{agg}}^{0}$ is the *K*_agg_ value measured in the presence of HSPB5 and in the absence of any crowders, $K_{\mathrm{agg}}^{1}$, $K_{\mathrm{agg}}^{2}$ and $K_{\mathrm{agg}}^{1,2}$ are the *K*_agg_ values measured in the presence of both HSPB5 and crowder 1, crowder 2, or their mixture, respectively. When the action of two crowding agents is independent, parameter *j* is equal to unity. When *j* > 1 there is a synergism in the combined action of two crowding agents, and the case *j* < 1 corresponds to the antagonism in their action. So slight antagonism was detected for the couple (PEG_20kDa_ + Ficoll_70kDa_). In contrast, synergism in the combined action was shown for pairs (PVP_10kDa_ + Ficoll_70kDa_; PVP_25kDa_ + Ficoll_70kDa_; PVP_10kDa_ + PEG_20kDa_; PVP_25kDa_ + PEG_20kDa_). The diagram in Appendix A illustrates graphically the antagonism of the action in the case of a mixture (PEG_20kDa_ + Ficoll_70kDa_) and the synergism for pairs (PVP_10kDa_ + Ficoll_70kDa_; PVP_25kDa_ + Ficoll_70kDa_; PVP_10kDa_ + PEG_20kDa_; PVP_25kDa_ + PEG_20kDa_) in the presence of HSPB5.

It can be seen from the diagram (Appendix A) that crowders can have different effects on *K*_agg_ in the absence of HSPB5. The presence of crowders in the absence of HSPB5 can not only increase the parameter *K*_agg_ (in the case of PVP_25kDa_; PVP_25kDa_ + PEG_20kDa_), which is the expected effect due to excluded volume, but also reduce *K*_agg_ (in the case of PEG; PVP_10kDa_; Ficoll_70kDa_; PVP_10kDa_ + Ficoll_70kDa_; PVP_25kDa_ + Ficoll_70kDa_; PVP_10kDa_ + PEG_20kDa_ and PEG_20kDa_ + Ficoll_70kDa_) (Appendix A, panel A). This deceleration of the Ph*b* aggregation can be explained by the action of three factors, such as: (1) an increase in the viscosity of the solution (see Appendix A), (2) the effect of crowding on the stage of denaturation of the Ph*b*, (3) weak interactions between the target protein and crowders. Appendix A (panel B) shows that nucleation time, *t*_0_, decreases both in the presence and in the absence of the chaperone. Thus, our data obtained by DLS method allow us to suggest that the chaperone-like activity of HSPB5 decreases under crowded conditions.

## 3. Discussion

It is well known that the dynamics and polydispersity of oligomers play an important role in the functioning of α-crystallins [10,29,90,91]. It was reported that subunit exchange promotes structural reorganization within the homo-oligomers of αB-crystallin [27] (see review [29]). The equilibrium between oligomeric forms is very sensitive to many factors such as temperature [1,27,92,93,94], post-translational modifications (phosphorylation) [33,35,95], divalent cations [31,32], crowding conditions [25], presence of target protein [25], and many others. Benesh and colleagues [90] suggested that the quaternary structure of α-crystallins is modulated by the assembly of oligomers from monomers or from dimers and there is an exchange between these forms, that have different conformation and chaperone-like activity [90]. Aquilina and coworkers showed that the population of αB-crystallin oligomers from the bovine eye lens contains oligomers consisting of an even and odd number of subunits [96]. They concluded that a monomer is the main building block of this assembly. Considering the monomers as the most active species, one can explain why αB-crystallin can prevent protein aggregation at very low stoichiometric ratio compared to the target protein. Such effective stoichiometry is possible in the cell, since the unfolded target proteins are often present at low concentrations and aggregate slowly (for example, in the lens). Therefore, a high concentration of active monomer species is not required to prevent the unfolding and aggregation of the target protein [29].

In the present work the stoichiometry of the HSPB5–target protein complex (*S*_0_) have been determined for two target proteins using Equation (3). In the case of heat-induced aggregation of Ph*b* we have shown that the dependence of the *K*_agg_/*K*_agg,0_ value on the ratio of molar concentrations of HSPB5 and Ph*b* ([HSPB5]/[target protein]) is complex, and the stoichiometry of the resulting chaperone–target protein complexes is variable (Figure 5). Possibly, the obtained complexity of the above dependence is associated with a change in the quaternary structure of HSPB5 with varying molar ratio [HSPB5]/[Ph*b*]. We suggest that the rather complicated dependence of the *K*_agg_/*K*_agg,0_ value on the molar ratio [HSPB5]/[Ph*b*] (Figure 5) is associated with: (1) the change in the oligomeric state of the chaperone with an increase in its concentration, and (2) possible variations in the affinity of HSPB5 species with respect to the denatured/aggregated protein. We assume that the lower the concentration of HSPB5, the greater is the fraction of “traveling monomers” in the solution at 48 °C [27] and, therefore, the more HSPB5 oligomers consist of monomeric building blocks [90]. With increasing concentration of HSPB5, the proportion of dimers and oligomers constructed from dimeric building blocks grows. The chaperone function of these two states is different: the monomeric substructural state has greater exposed hydrophobic surface area and is consequently more active [90]. All the above mentioned is likely to underlie the obtained value of the stoichiometry of the chaperone–target protein complexes at the initial stages of Ph*b* aggregation at 48 °C at low ratios of molar concentrations [HSPB5]/[Ph*b*] (in the excess of the molar concentration of Ph*b*): when one monomer of HSPB5 (with the molecular mass of 20.2 kDa) is complexed with over three Ph*b* monomers (with the molecular mass of 97.4 kDa each).

When studying the kinetics of DTT-induced aggregation of αLa (a protein with a molecular mass of the monomer of 14.2 kDa, smaller than that of HSPB5 subunit) in the presence of HSPB5 (Figure 7), we obtained an even lower value of the stoichiometry of the chaperone–target protein complexes, namely less than 0.011. The oligomeric state of HSPB5–target protein complexes was controlled by the AUC method. According to the SV analysis we conclude that suboligomeric forms of HSPB5 (monomeric or dimeric) may interact with the target protein, DTT-denatured αLa, forming relatively small oligomers (see Table 1). However, the high-order oligomeric HSPB5–αLa complexes may be present. According to Hayashi and Carver the monomeric form of HSPB5 may be its most chaperone-species active [29]. However, the presence of monomers can increase the hydrophobicity of HSPB5 and decrease its solubility. Therefore, for the chaperone-like activity, the balance between monomeric and oligomeric forms is very important.

Previously, we showed a significant effect of crowding on the oligomeric state of HSPB5 under conditions of the heat shock (48 °C) by the SV method [25]. It was demonstrated that, on the one hand, an elevated temperature led to the dissociation of the chaperone, and, on the other hand, crowding shifted the equilibrium toward large-sized assemblies [25]. Based on this data we assumed that the chaperone-like activity of HSPB5 is likely to be affected by the crowding.

In the present work, to quantify the effect of crowding on the chaperone-like activity of HSPB5, we selected parameter *K*_agg_ characterizing the acceleration of the aggregation process during the nucleation stage. Ph*b* aggregation at 48 °C was chosen as a test system. Table 2 and Appendix A show the effect of crowding on the kinetic parameters of Ph*b* thermal aggregation. When considering the effect of crowding on Ph*b* aggregation in the presence of HSPB5, it must be emphasized that all crowders used, as well as their pairs, increase the value of parameter *K*_agg_ as compared to that in the buffer, although to a different extent. That is, the aggregation of the target protein in the presence of HSPB5 under the conditions of crowding created by all used crowders or their mixtures is accelerated compared to the aggregation process in the buffer. Thus, the data obtained by the DLS method suppose that crowding reduces the chaperone-like activity of HSPB5.

It may be assumed that crowding accelerates the aggregation of the chaperone–target protein complexes. However, the data obtained by the SV method indicate the retention of complexes formed by the dissociated forms of HSPB5 and Ph*b* under crowded conditions (Figure 10 and Figure 11). This is evidenced by the disappearance of the 23 S peak and a decrease in the 9 S peak in the *c*(*s*) distributions corresponding to the sedimentation of individual HSPB5 and Ph*b*, respectively, and the appearance of peaks with *s*_20,w_ values of 11, 14, 16 S for HSPB5–Ph*b* mixture under crowded conditions arising from the presence of the pair of crowders, PVP_25kDa_ + PEG_20kDa_ (Figure 11). That is, under mixed crowding conditions, although a 55-fold increase in the value of parameter *K*_agg_ was registered by DLS (compared with the buffer, Table 2), complexes formed by the suboligomeric forms of HSPB5 with the target protein are retained in the solution. This fact is consistent with our earlier data on the interaction of HSPB5 with Ph*b* at 48 °C in the presence of the pair of crowders, PEG_20kDa_ and TMAO [25]. The presence of a target protein may stimulate the dissociation of large HSPB5 assemblies. However, the existence of high-order oligomeric HSPB5–Ph*b* complexes cannot be ruled out. The obtained results support our previous data on the formation of the complexes between dissociated forms of bovine lens α-crystallin and an apoform of Ph*b* [63], or Ph*b* denatured by ultraviolet radiation [62], apart from the high order complexes. The presence of two types of complexes formed by α-crystallin and target proteins, which differ in their sensitivity to crowding and aggregation, was reported in the work [62]. High molecular mass complexes are aggregation-prone, whereas complexes formed by small suboligomeric forms of chaperone with a target protein are more resistant to aggregation under crowding conditions [36,62]. We suggested that these small complexes are responsible for the realization of the chaperone-like activity of HSPB5 or α-crystallin under crowded conditions [25,36,62]. Our results are consistent with this idea.

Thus, we showed that by changing the combination of different crowding agents, almost spherical crowders, like Ficoll, or linear polymers, like PVP, it is possible to regulate (moderate) the activity of the chaperone. Our studies show that adding even one crowder to an existing one dramatically changes the effectiveness of the crowding. Parameter *K*_agg_ increases in 13.7–55 times in the presence of those pairs of crowders that exhibit synergism (Table 2). On the one hand, this can be explained by the increase (strengthening) of the excluded volume effect, which leads to acceleration of the target protein aggregation. On the other hand, it is currently believed that, in addition to the excluded volume effect (steric repulsion of macromolecules – the entropy factor), the forces of weak interaction between protein molecules and crowder molecules (enthalpy factor) play a significant role in crowded environment [49,50,97,98,99,100]. In addition, biopolymers (proteins) capable of reversibly changing the state of association/dissociation or accepting an expanded or compact state of the quaternary structure can change the level of excluded volume in cells [101,102]. It is assumed that proteins with such properties, through their ability to directly influence the degree of excluded volume, will dynamically regulate the functions of proteins in biological media [102]. Small heat shock proteins, such as HSPB5, can take extended or compact states [9,20] and change the state of association, therefore may affect the effective level of excluded volume. Thus, crowding adds another level of complexity to the relationship between the activity and structural dynamics of sHSPs.

An increase in the concentration of sHSPs and their activity during a stress play a protective role in the cell, but there are situations when the activity of the chaperone should be reduced, for example, in cancer cells. Crowding should be considered as one of the factors that may subtly modulate sHSPs activity.

## 4. Materials and Methods

### 4.1. Materials

Ca^2+^-depleted α-lactalbumin (αLa), mono- and dibasic sodium phosphates, Hepes, ethylenediaminetetraacetic acid (EDTA), Ficoll with a molecular mass of 70,000 Da (Ficoll_70kDa_), polyvinylpyrrolidone (PVP) with a molecular mass of 10,000 Da (PVP_10kDa_) and 25,000–30,000 Da (PVP_25kDa_) were purchased from Sigma-Aldrich (St. Louis, MO, USA). 1,4-dithiothreitol (DTT) was purchased from PanReac (Barcelona, Spain). Polyethylene glycol (PEG) with a molecular mass of 20,000 Da (PEG_20kDa_) was purchased from Ferak Berlin (Berlin, Germany). NaCl was purchased from Reakhim (Moscow, Russia). All solutions for the experiments were prepared using deionized milli-Q quality water obtained with the Easy-Pure II RF system (Thermo Fisher Scientific, Barnstead, city, NH, USA). The procedure of isolation of glycogen phosphorylase *b* (Ph*b*, Uniprot IDP00489) from rabbit skeletal muscles was described in the work [103]. Human full-length untagged HSPB5 (Uniprot ID P02511) was expressed and purified as described earlier [104] and stored frozen at −20 °C. Size-exclusion chromatography elution profile of HSPB5 on a Superdex 200 column and SDS-gel electrophoresis of the stock HSPB5 sample and its SEC elution fractions are presented in Appendix A. The authors thank Nikolai Sluchanko for a gift of the HSPB5 preparation and its characteristics.

The αLa solutions were prepared by dissolving the desired amount of lyophilized protein powder in 0.1 M Na-phosphate buffer, pH 6.8. Before DLS measurements the solutions were centrifuged at 20,000× *g* and +4 °C for 30 min to clear them from insoluble particles. Stock Ph*b* and HSPB5 solutions were centrifuged at 12,850× *g* and +4 °C for 10 min after the dialysis against 30 mM Hepes buffer, pH 6.8, containing 0.1 M NaCl, 0.2 mM EDTA. Protein concentrations were determined using a Nanophotometer P330 (Implen GmbH, Munich city, Germany) with the extinction coefficient $\varepsilon_{280}^{0.1\%}$ equal to 1.32 for Ph*b* [105], 0.693 for HSPB5 [104] and 2.01 for αLa [106].

### 4.2. Dynamic Light Scattering (DLS)

Aggregation of the proteins under the study was monitored by changes in the light scattering intensity at 90° angle. The light scattering kinetic curves were registered by Photocor Complex correlation spectrophotometer (PhotoCor Instruments, Inc., College Park, MD, USA) with a He-Ne laser (Coherent Inc., Santa Clara, CA, USA, Model 31-2082, 632.8 nm, 10 mW) as a light source. All experiments were repeated at least three times.

The kinetics of thermal aggregation of Ph*b* in the absence and in the presence of HSPB5 was studied both in dilute (30 mM Hepes buffer, pH 6.8, containing 0.1 M NaCl, 0.2 mM EDTA) and crowded solutions (the same buffer with the addition of Ficoll_70kDa_, PEG_20kDa_, PVP_10kDa_, PVP_25kDa_, or mixtures of these crowding agents) at 48 °C. All solutions were prepared on the buffer passed through a 20 nm “Anotop” filter (Whatman, Cytiva, Little Chalfont, Buckinghamshire, UK). The aggregation process was initiated by the addition of an aliquot of Ph*b* to the solution (lacking/containing HSPB5 and/or crowding agents) pre-incubated at 48 °C. The final volume of the probe was 0.5 mL. Cells with a stopper were used to avoid evaporation during the experiment. The diffusion coefficient (*D*), measured at 48 C, were estimated based on 5 independent measurements.

It should be noted that the absolute values of the parameters *K*_agg_ and *t*_0_ calculated from Equation (1) may vary in the sets of experiments conducted with protein samples of different extractions (owing to circumstances beyond our control). The experience of our many years of work with Ph*b* shows that the preparation of freshly isolated Ph*b* is relatively stable for about two weeks after isolation. Nevertheless, the properties of Ph*b* (kinetic parameters of the enzymatic reaction, thermal stability, oligomeric state and other properties) undergo small changes during about two weeks after isolation. This means that each series of experiments is being carried out for one day and the necessary comparisons can be made within this series.

To register the kinetic curves of DTT-induced αLa aggregation in the absence and in the presence of HSPB5, the aliquots of the chaperone and αLa were placed in a buffer solution (0.1 M Na-phosphate buffer, pH 6.8) pre-heated at 37 °C to obtain the desired final concentrations. The aggregation process was initiated by the addition of DTT to the final concentration of 20 mM.

DynaLS software (Alango, Tirat Carmel, Israel) was used for polydisperse analysis of DLS data. The values of the refractive index and the dynamic viscosity of the solutions used in the DLS measurements at 48 °C are presented in Appendix A.

### 4.3. Analytical Ultracentrifugation (AUC)

Sedimentation velocity (SV) experiments were carried out using a Model E analytical ultracentrifuge (Beckman Instruments, Palo Alto, CA, USA) equipped with absorbance optics, a photoelectric scanner, a monochromator and a computer on-line. In the case of Ph*b* SV runs were performed in 30 mM Hepes buffer, pH 6.8, containing 0.1 M NaCl, 0.2 mM EDTA, with or without different crowding agents at 48 °C. Before the experiment, all samples were incubated for 40 min at 48 °C. A four-hole rotor An-F Ti and 12-mm double sector cells were used. Before the run, the rotor was pre-heated in a thermostat at 48 °C overnight. In the case of αLa, SV runs were performed in 0.1 M Na-phosphate buffer, pH 6.8, containing 10 mM NaCl and 20 mM DTT, at 37 °C. Before the run, the rotor An-F Ti was pre-heated in a thermostat at 37 °C overnight. Sedimentation profiles were recorded by measuring the absorbance at 280 nm. All cells were scanned simultaneously. The time interval between scans was 2.5 min. The differential sedimentation coefficient distributions [*c*(*s*) versus *s*] were determined using SEDFIT program [107]. Sedimentation coefficients were corrected to the standard conditions (a solvent with the density and viscosity of water at 20 °C) using SEDFIT [107]. The values of the density and the dynamic viscosity of the solutions used in the AUC measurements at 48 °C are presented in Appendix A.

To estimate the molecular mass of Ph*b* and possible complexes of Ph*b* with HSPB5, the Svedberg equation was used: *M* = *sR*T/*D*(1 − *νρ*),(6)
where *ν* is the partial specific volume of a protein, *ρ* is solution density, *R* is molar gas constant, *T* is temperature in Kelvin, *s* is a sedimentation coefficient, *D* is a diffusion coefficient. For calculations we used sedimentation coefficients (*s*), determined by AUC, and diffusion coefficients (*D*), determined by DLS at 48 °C.

### 4.4. Calculations

The Origin Pro 2017 (Origin Lab Corp., Northampton, MA, USA) software was used for the calculations. The *R*^2^ coefficient of determination was used in order to characterize the degree of agreement between the experimental data and calculated values [108].

## Figures and Tables

**Figure 1 ijms-21-04940-f001:**
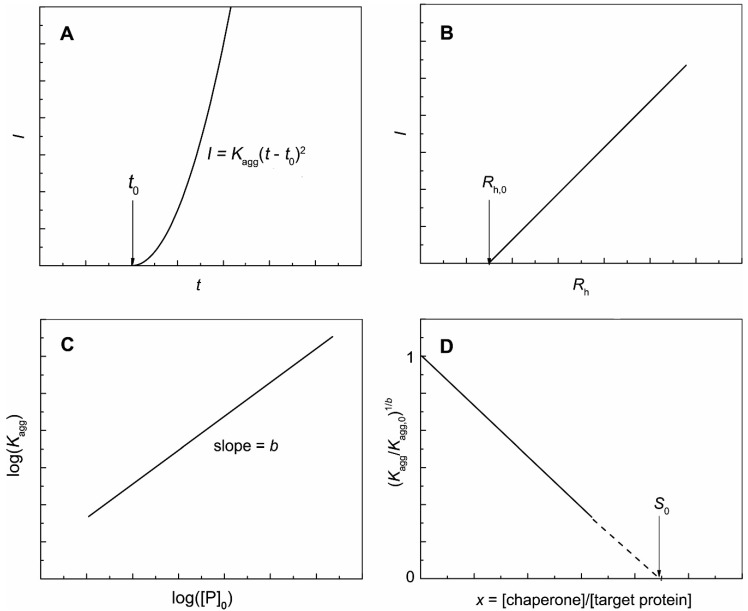
Theoretical analysis of the initial parts of the kinetic curves registered for protein aggregation by measuring the increment of the light scattering intensity. (**A**) The dependence of the light scattering intensity (*I*) on time (*t*) obeying Equation (1). (**B**) The light scattering intensity (*I*) versus the hydrodynamic radius (*R*_h_) plot used for calculation of the hydrodynamic radius of start aggregates (*R*_h,0_). (**C**) The log(*K*_agg_) versus log([P]_0_) plot used for calculation of the parameter *b*. (**D**) The (*K*_agg_/*K*_agg,0_)^1/*b*^ versus *x* = [chaperone]/[target protein] plot used for calculation of the stoichiometry of the chaperone–target protein complex (*S*_0_).

**Figure 2 ijms-21-04940-f002:**
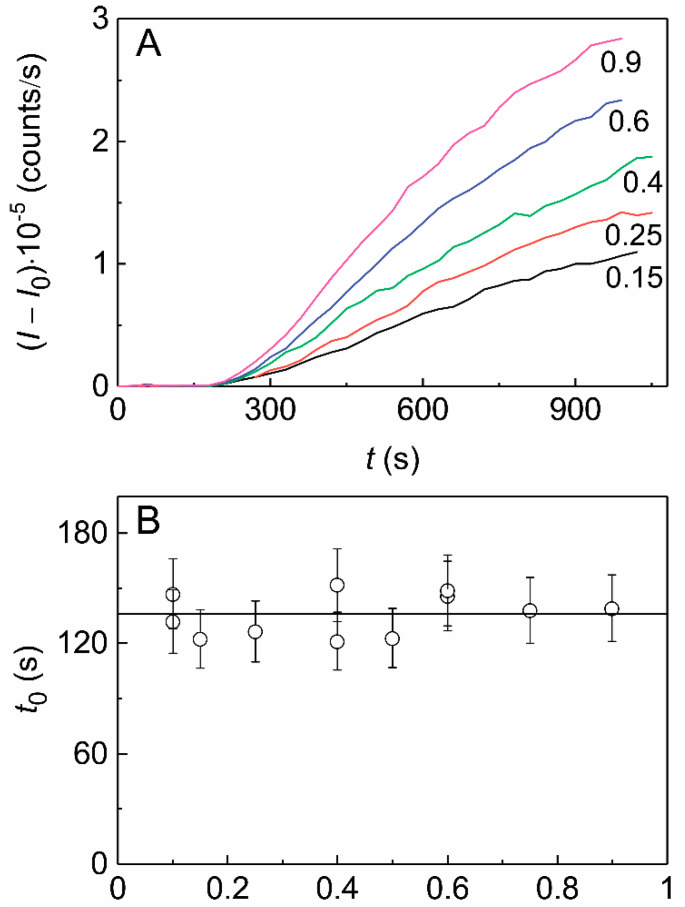
The kinetics of thermal aggregation of Ph*b* at 48 °C. (**A**)The dependences of the light scattering intensity (*I–I*_0_) on time (*t*) for aggregation of Ph*b* at different protein concentrations (figures near the curves correspond to Ph*b* concentration in mg/mL). (**B**,**C**) The dependences of the parameters *t*_0_ and *K*_agg_, respectively, calculated according to the Equation (1), on Ph*b* concentration. The solid lines in panels (**B**,**C**) are the linear approximations of the experimental data.

**Figure 3 ijms-21-04940-f003:**
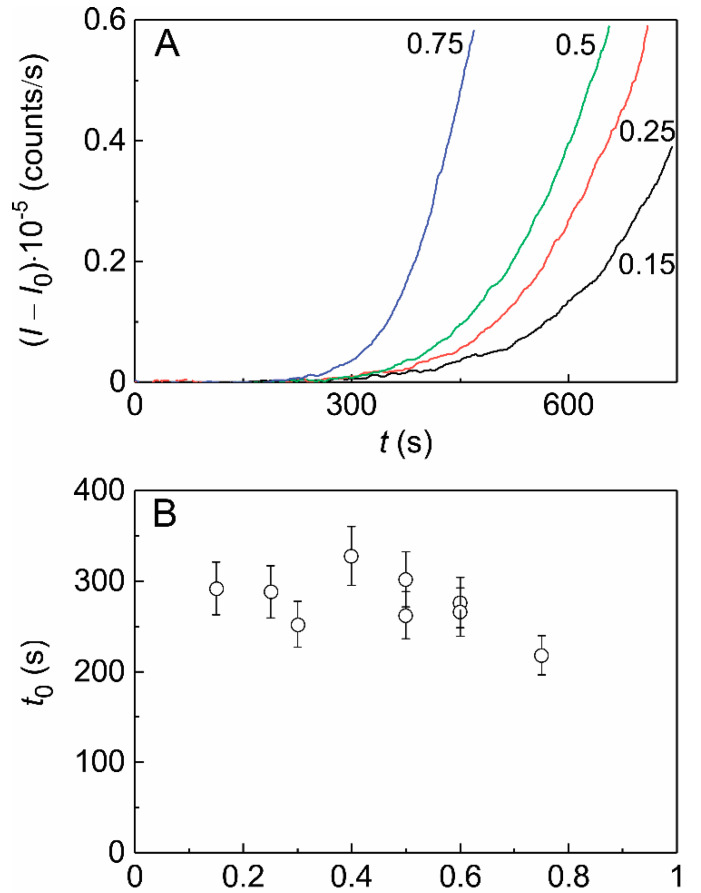
The kinetics of thermal aggregation of Ph*b* at 48 °C registered in the presence of HSPB5. (**A**) The initial parts of the dependences of the light scattering intensity (*I–I*_0_) on time (*t*) obtained for Ph*b* aggregation in the presence of HSPB5 (0.025 mg/mL) and various concentrations of Ph*b*. Figures near the curves correspond to Ph*b* concentration in mg/mL. (**B**) The dependence of parameter *t*_0_ on Ph*b* concentration. (**C**) The dependence of parameter *K*_agg_ on Ph*b* concentration. The solid line is the linear approximation of the experimental data. The dashed line corresponds to the linear approximation of the dependence of *K*_agg_ on [Ph*b*] for aggregation of Ph*b* in the absence of HSPB5.

**Figure 4 ijms-21-04940-f004:**
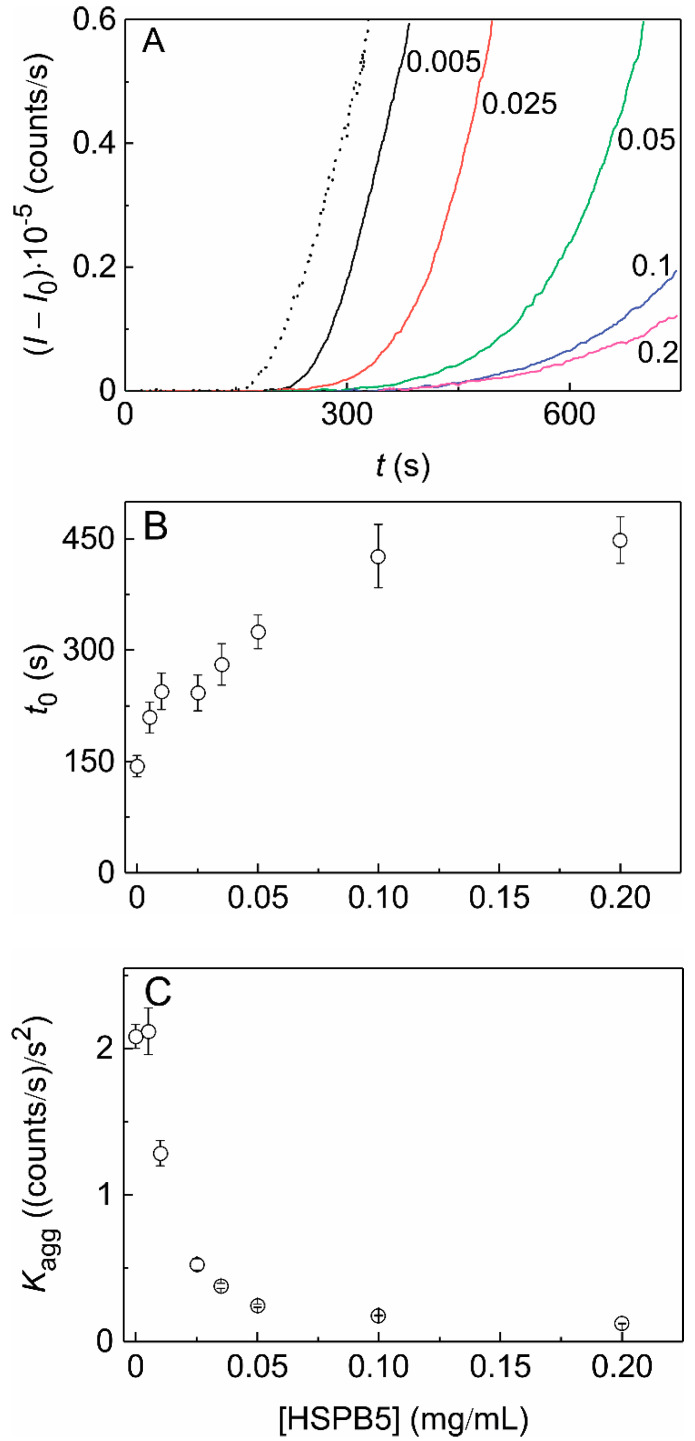
The effect of HSPB5 on the kinetics of thermal aggregation of Ph*b* at 48 °C. (**A**) The initial parts of the dependences of the light scattering intensity (*I–I*_0_) on time (*t*) obtained for Ph*b* aggregation ([Ph*b*] = 0.4 mg/mL) at various concentrations of HSPB5. Figures near the curves correspond to HSPB5 concentration in mg/mL. The dotted line corresponds to the aggregation of Ph*b* alone. (**B**,**C**) The dependence of parameter *t*_0_ on the concentration of HSPB5 and the dependence of parameter *K*_agg_ on the concentration of HSPB5, respectively.

**Figure 5 ijms-21-04940-f005:**
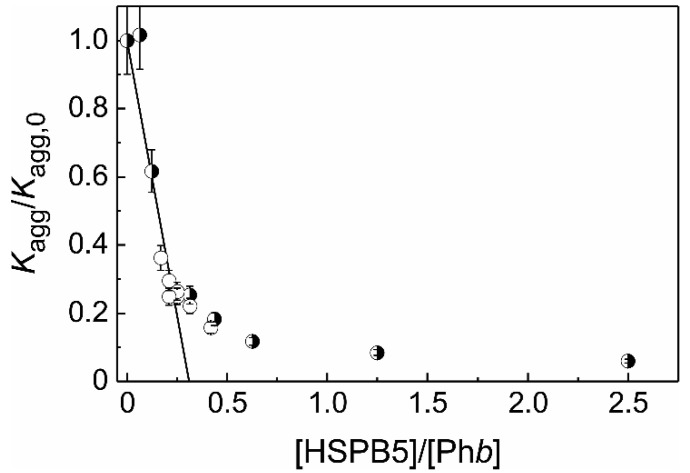
The suppression of Ph*b* aggregation by HSPB5. The dependence of *K*_agg_/*K*_agg,0_ ratio on the ratio of molar concentrations of HSPB5 and Ph*b* (*x* = [HSPB5]/[Ph*b*]). Open circles correspond to *K*_agg_/*K*_agg,0_ values at constant HSPB5 concentration (0.025 mg/mL). Half-filled circles correspond to *K*_agg_/*K*_agg,0_ values at constant Ph*b* concentration (0.4 mg/mL). The solid line is calculated from the equation *K*_agg_/*K*_agg,0_ = 1 − *x*/*S*_0_ at *S*_0_ = 0.31.

**Figure 6 ijms-21-04940-f006:**
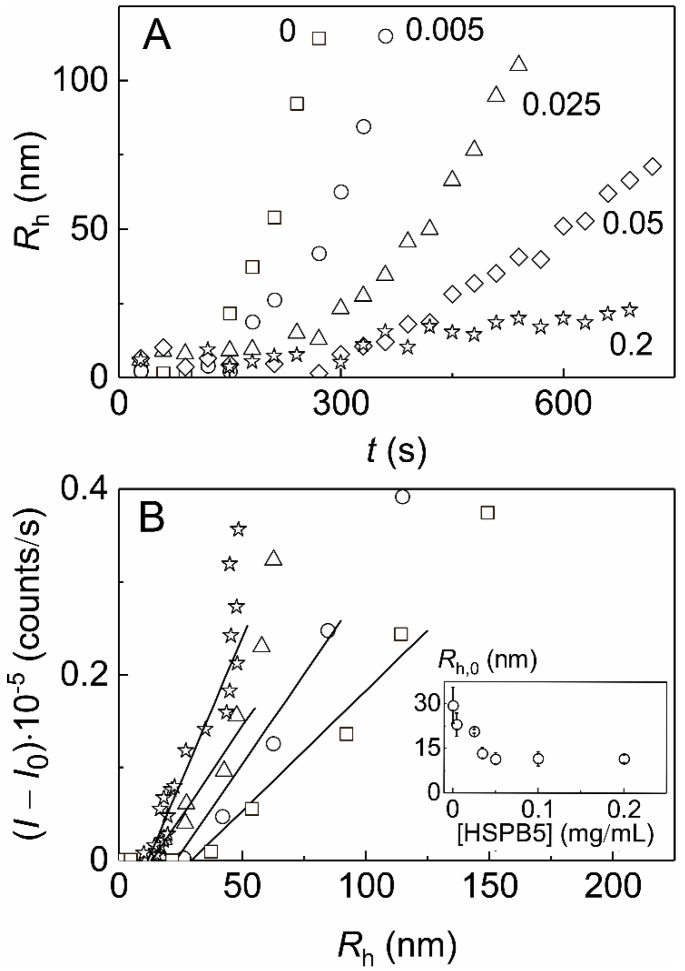
The effect of HSPB5on the thermal aggregation of Ph*b* at 48 °C. (**A**) The dependences of the hydrodynamic radius (*R*_h_) of Ph*b* aggregates on time (*t*) obtained at various concentrations of HSPB5 ([Ph*b*] = 0.4 mg/mL; figures near the curves correspond to HSPB5 concentration in mg/mL). (**B**) The relationship between the light scattering intensity (*I–I*_0_) and the *R*_h_ values of Ph*b* aggregates in the absence of HSPB5 (squares) and in the presence of HSPB5 at the following concentrations: 0.005 (circles), 0.035 (triangles) and 0.2 mg/mL (stars). The solid lines are the linear approximations of the experimental data. The inset shows the dependence of the hydrodynamic radius of start aggregates (*R*_h,0_) on HSPB5 concentration.

**Figure 7 ijms-21-04940-f007:**
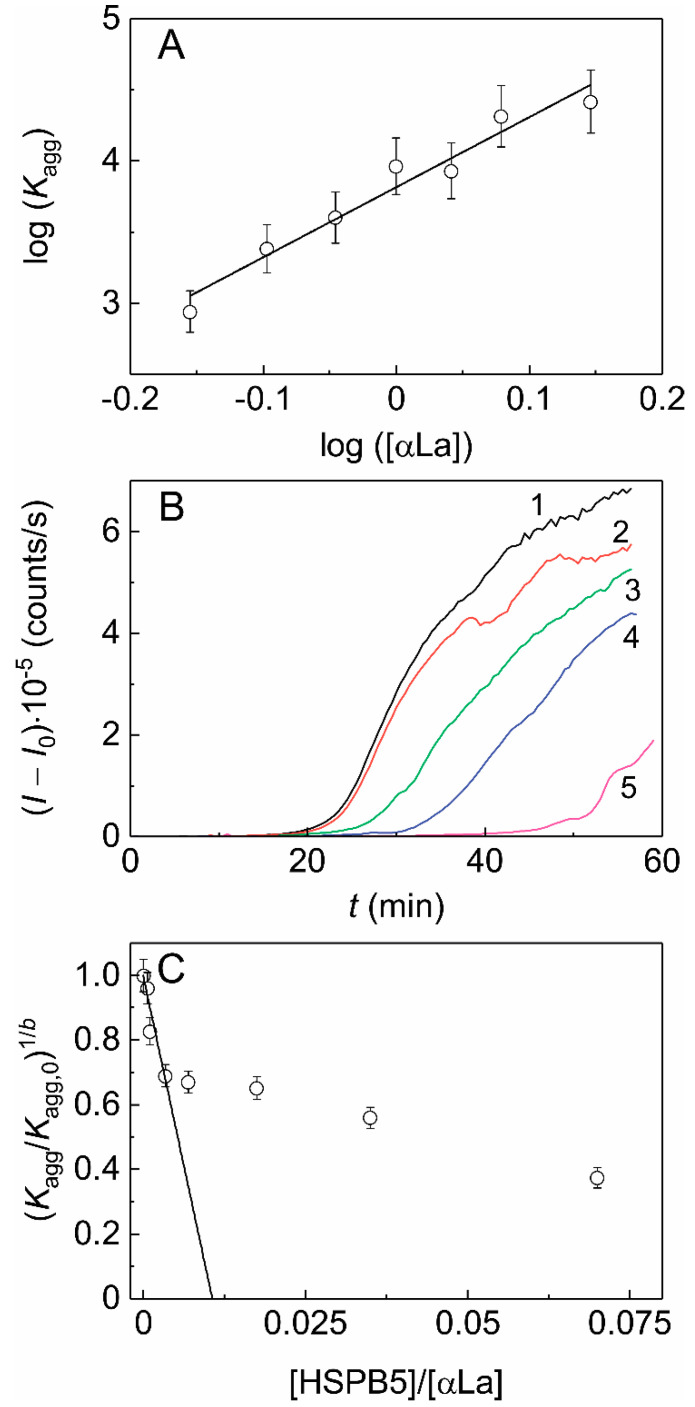
The effect of HSPB5 on DTT-induced aggregation of α-lactalbumin (αLa). (**A**) The relationship between parameter *K*_agg_ and the initial concentration of αLa in coordinates {log([αLa]); log(*K*_agg_)}. The slope of the linear fitting is equal to parameter *b* (*b* = 4.9 ± 0.5). (**B**) The dependences of the light scattering intensity (*I*) on time for αLa aggregation ([αLa] = 1 mg/mL, 0.1 M Na-phosphate buffer, pH 6.8, 20 mM DTT, 37 °C) in the absence of HSPB5 (curve1) and in the presence of the following concentrations of HSPB5: (2) 0.001, (3) 0.0025, (4) 0.0075 and (5) 0.025 mg/mL. (**C**) The dependence of the relative parameter (*K*_agg_/*K*_agg,0_)^1/*b*^ on the ratio of the molar concentrations of the chaperone and the target protein (*x*). The linear fitting is drawn according to Equation (3).

**Figure 8 ijms-21-04940-f008:**
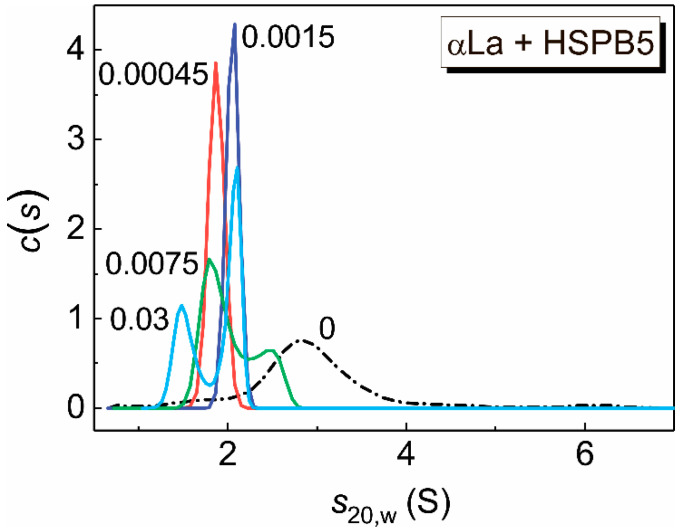
The sedimentation behavior of αLa denatured by the addition of 20 mM DTT in the presence of HSPB5 at 37 °C. The differential sedimentation coefficient distributions, *c*(*s*), for αLa (0.3 mg/mL) at various concentrations of HSPB5. Figures near the curves correspond to HSPB5 concentration in mg/mL. The *c*(*s*) distributions were transformed to standard conditions. Rotor speed was 60,000 rpm. The total time of the experiment at the elevated temperature was 140 min.

**Figure 9 ijms-21-04940-f009:**
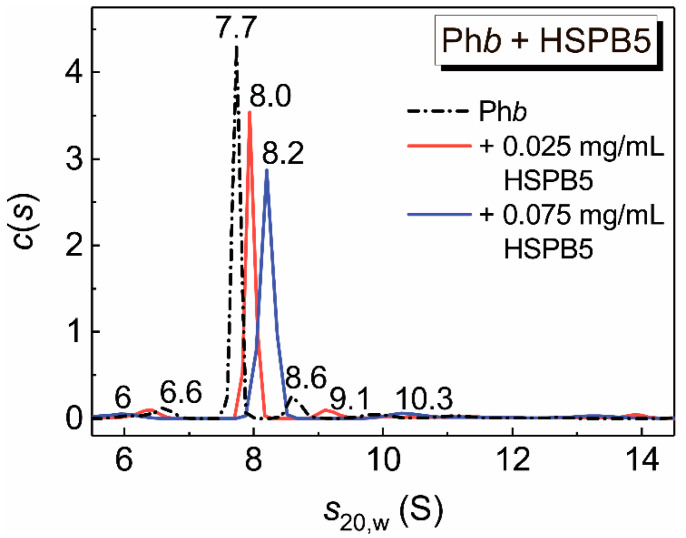
Interaction of Ph*b* with HSPB5 at 48 °C. (A) Differential distributions *c*(*s*) for Ph*b* alone (0.5 mg/mL; dash dot line) and the mixtures of Ph*b* (0.5 mg/mL) and HSPB5 at the following concentrations: 0.025 mg/mL (red) and 0.075 mg/mL (blue). The *c*(*s*) distributions were corrected to the standard conditions. Rotor speed was 48,000 rpm. The total time of the experiment at the elevated temperature was 90 min.

**Figure 10 ijms-21-04940-f010:**
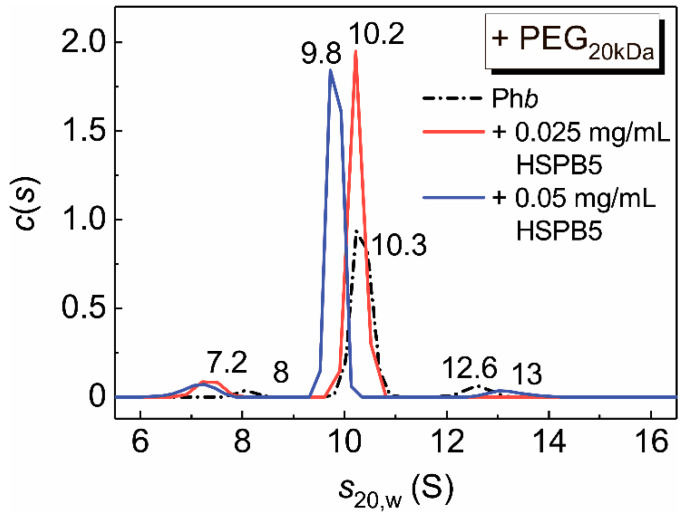
The *c*(*s*) distributions for the mixtures of Ph*b* (0.5 mg/mL) and HSPB5 in the presence of PEG_20kDa_ (25 mg/mL). The following concentrations of HSPB5 were used: 0 (black), 0.025 mg/mL (red) and 0.05 mg/mL (blue). The *c*(*s*) distributions were corrected to the standard conditions. Rotor speed was 48,000 rpm. The total time of the experiment at the elevated temperature was 90 min.

**Figure 11 ijms-21-04940-f011:**
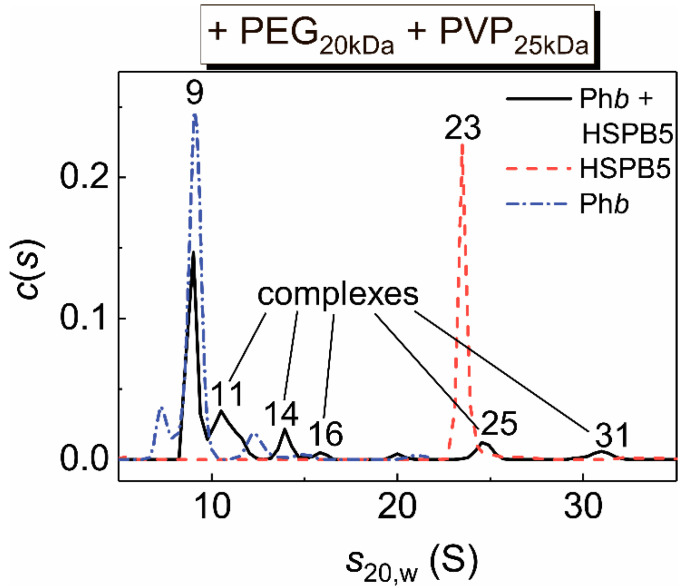
Interaction of HSPB5 with Ph*b* as a target protein at 48 °C under crowded conditions arising from the presence of the mixture of PEG_20kDa_ (25 mg/mL) and PVP_25kDa_ (12.5 mg/mL). The differential sedimentation coefficient distributions, *c*(*s*), for Ph*b* (0.4 mg/mL; dash dot blue line), HSPB5 (0.26 mg/mL; dash red line) and its mixture (solid black line) are presented. The *c*(*s*) distributions were obtained at 48 °C and transformed to standard *s*_20,w_ distributions. Rotor speed was 48,000 rpm. The total time of the sedimentation experiment at the elevated temperature was about 90 min.

**Table 1 ijms-21-04940-t001:** Estimation of the molar mass and *s*_20,w_ values of the complexes between HSPB5 and αLa denatured by 20 mM DTT (0.1 M Na-phosphate buffer, 0.01 M NaCl, pH 6.8, 37 °C).

[HSPB5] (mg/mL)	*s*_20,w_ (S)	Friction Ratio, *f/f*_0_	Molecular Mass (kDa)
0	2.9 ± 0.8 6.1 ± 0.4	2.7	
0.00045	1.85 ± 0.13	1.614	25.4
0.0015	2.0 ± 0.1	1.545	26
0.0075	1.9 ± 0.2 2.4 ± 0.3	1.925	33 47
0.03	1.6 ± 0.2 2.0 ± 0.1	1.836	23.9 37.4

**Table 2 ijms-21-04940-t002:** Kinetic parameters for aggregation of Ph*b* (0.4 mg/mL) in the presence of HSPB5 at a concentration of 0.025 mg/mL at 48 °C (0.03 M Hepes buffer, 0.1 M NaCl, 0.2 mM EDTA, pH 6.8).

Additions	*K*_agg_ ((counts/s)/s^2^)	*t*_0_ (s)	$K_{\mathrm{agg}}/K_{\mathrm{agg}}^{0}$ ^1^	*j*
Without addition of crowding agents				
–	0.123 ± 0.007	323 ± 9	1.0	-
Action of individual crowding agents				
PEG_20kDa_ 25 mg/mL	1.85 ± 0.04	287 ± 2	15.0 ± 0.9	-
PVP_10kDa_ 25 mg/mL	0.28 ± 0.01	266 ± 5	2.3 ± 0.2	-
PVP_25kDa_ 25 mg/mL	0.52 ± 0.02	232 ± 5	4.2 ± 0.3	-
Ficoll_70kDa_ 75 mg/mL	0.144 ± 0.015	252 ± 12	1.17 ± 0.14	-
Combined action of crowding agents				
PEG20kDa 25 mg/mL + Ficoll70kDa 75 mg/mL	1.37 ± 0.05	196 ± 4	11.1 ± 0.8	0.71 ± 0.06
PVP_10kDa_ 25 mg/mL + Ficoll_70kDa_ 75 mg/mL	1.69 ± 0.05	263 ± 3	13.7 ± 0.9	8.8 ± 0.6
PVP_25kDa_ 25 mg/mL + Ficoll_70kDa_ 75 mg/mL	3.76 ± 0.21	240 ± 3	30.6 ± 2.4	8.7 ± 0.7
PVP_10kDa_ 25 mg/mL + PEG_20kDa_ 25 mg/mL	4.00 ± 0.22	227 ± 3	32.5 ± 2.6	2.06 ± 0.17
PVP_25kDa_ 25 mg/mL + PEG_20kDa_ 25 mg/mL	6.76 ± 0.29	203 ± 2	55 ± 4	3.12 ± 0.23

^1^$K_{\mathrm{agg}}^{0}$ is the *K*_agg_ value measured in the presence of HSPB5 and in the absence of any crowders.

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
