# Peer review of "Chaperone-Like Activity of HSPB5: The Effects of Quaternary Structure Dynamics and Crowding"

_ijms, 2020, doi:10.3390/ijms21144940_

Round 1
Reviewer 1 Report
The authors have used the quadratic equation to estimate the parameters of protein aggregation. They have looked at the effect of molecular crowding on the chaperone-like function of HSPB-5. The study is exciting and adds to our understanding of the functioning of the chaperone protein.
The following comments might help improve the manuscript.
Will the t0 and Kagg values for a given substrate protein concentration change if the experiments are repeated at different times? In Fig 3, when the aggregation performed using 0.4 mg/mL of Phb and 0.025 mg/mL HSPB5, the t0 value is found to be above 300 s, and the Kagg value is below 0.25. In Fig 4, at the same concentration of substrate and chaperone proteins, the t0 value reported is below 250 s, and the Kagg value is about 0.5. Also, in Fig 2, the Kagg value for 0.4 mg/ml Phb is 1.0, and in Fig S2 for the same concentration of Phb, the Kagg value is reported as 2.0. The authors should clarify why the values are different for the same substrate concentration in various experiments. The AC0 values of the chaperone protein determined using the formula may not be accurate if the Kagg value of the substrate is changing.
The AC0 value calculated from the aggregation kinetics data suggests the binding of 91 aLa molecules per one HSPB5 subunit. However, the estimation of molar mass (Table 1) does not indicate the formation of such large complexes. Did the complexes precipitate during analysis? Why was the rotor speed set at 60,000 rpm (Fig 8) while the remaining studies were performed at 48,000 rpm?
How were the samples processed for DLS analysis? Where the samples centrifuged or filtered before analysis?
Where the precipitated samples excluded from SV analysis?
Give the estimated molar mass values of the complexes between HSPB5 and Phb (Fig 9).
Explain why the S20,W values for Phb with HSPB5 increase (Figure 9), while the Rh values show an apparent reduction in the size of Phb in the presence of HSPB5 (Figure 6).
Can the fractions of SV analysis collected and run on SDS-PAGE gel or analyzed by Western blot analysis to validate the claims made in the paper? It would be helpful if Rh and Molar mass of the SV peaks is determined.
The crowding effect on the HSPB5 chaperone function should also be tested with the Lysozyme substrate to see if the observed effect is substrate-dependent.
Words are merged in many places, especially in the introduction section.
Line 35 change “apaptosis” to “apoptosis.”
Author Response
Manuscript ID: ijms-843396
Title: Chaperone-like activity of HSPB5: the effects of quaternary structure dynamics and crowding
Natalia A. Chebotareva*, Svetlana G. Roman, Vera A. Borzova, Tatiana B. Eronina, Valeriya V. Mikhaylova and Boris I. Kurganov
Dear Ms. Jade Yu
Thank you very much for your letter of 2020-25-06 concerning our paper “Chaperone-like activity of HSPB5: the effects of quaternary structure dynamics and crowding”. We are thankful to you and the reviewers for the valuable comments. We have taken into account all the comments.
We send you the revised manuscript.
Reviewer #1
Dear Reviewer,
Thank you very much for your revision concerning our paper “Chaperone-like activity of HSPB5: the effects of quaternary structure dynamics and crowding”. We are thankful to you for the valuable comments took them into account by correcting our article.
Reviewers' comments:
The authors have used the quadratic equation to estimate the parameters of protein aggregation. They have looked at the effect of molecular crowding on the chaperone-like function of HSPB-5. The study is exciting and adds to our understanding of the functioning of the chaperone protein.
The following comments might help improve the manuscript.
Comment 1
Will the t0 and Kagg values for a given substrate protein concentration change if the experiments are repeated at different times? In Fig 3, when the aggregation performed using 0.4 mg/mL of Phb and 0.025 mg/mL HSPB5, the t0 value is found to be above 300 s, and the Kagg value is below 0.25. In Fig 4, at the same concentration of substrate and chaperone proteins, the t0 value reported is below 250 s, and the Kagg value is about 0.5. Also, in Fig 2, the Kagg value for 0.4 mg/ml Phb is 1.0, and in Fig S2 for the same concentration of Phb, the Kagg value is reported as 2.0. The authors should clarify why the values are different for the same substrate concentration in various experiments. The AC0 values of the chaperone protein determined using the formula may not be accurate if the Kagg value of the substrate is changing
Reply
The absolute values of the parameters Kagg and t0 may vary in the sets of experiments conducted with protein samples of different extractions (owing to circumstances beyond our control). The experience of our many years of work with Phb (starting from works Silonova G.V., Livanova N.B., Kurganov B.I., Allosteric inhibition of phosphorylase b from rabbit skeletal muscles, Molek. Biol. (Mosk), 1969, 3, 768; Silonova G.V., Kurganov B.I., Studies of rabbit muscle phosphorylase b association by a kinetic method, Molek. Biol. (Mosk), 1970, 4, 445.) shows that the preparation of freshly isolated Phb is relatively stable for about two weeks after isolation. Nevertheless, the properties of Phb (kinetic parameters of the enzymatic reaction, thermal stability, oligomeric state and other properties) undergo small changes during about two weeks after isolation. This means that each series of experiments is being carried out for one day and the necessary comparisons can be made within this series.
That is why the relative value of the parameter Kagg (Kagg/Kagg,0, where Kagg,0 is the value of Kagg obtained for aggregation of Phb in the absence of the SHSP, a kind of a “control” value for each experiment) rather than its absolute value was chosen for our consideration in order to make it possible to use and compare the results of different experimental sets. As we can see in Figure 5, data from different experiments fall on one curve, making it reasonable to determine the apparent adsorption capacity of the chaperone from the obtained dependence of the Kagg/Kagg,0 on the [HSPB5]/[Phb] ratio.
The following text has been inserted on page 19 of the manuscript at the end of the Section 4.2. Dynamic Light Scattering (DLS):
It should be noted that the absolute values of the parameters Kagg and t0 calculated from Equation (1) may vary in the sets of experiments conducted with protein samples of different extractions (owing to circumstances beyond our control). The experience of our many years of work with Phb shows that the preparation of freshly isolated Phb is relatively stable for about two weeks after isolation. Nevertheless, the properties of Phb (kinetic parameters of the enzymatic reaction, thermal stability, oligomeric state and other properties) undergo small changes during about two weeks after isolation. This means that each series of experiments is being carried out for one day and the necessary comparisons can be made within this series.
Comment 2
The AC0 value calculated from the aggregation kinetics data suggests the binding of 91 aLa molecules per one HSPB5 subunit. However, the estimation of molar mass (Table 1) does not indicate the formation of such large complexes. Did the complexes precipitate during analysis?
Why was the rotor speed set at 60,000 rpm (Fig 8) while the remaining studies were performed at 48,000 rpm?
Reply
It was reported that native mass spectrometry method revealed an increase in the number of αB-crystallin monomers with increased temperature, which was accompanied by a concomitant increase in the rate of subunit exchange. An increase in the rate of subunit exchange can release more monomers, allowing for a greater number of molecules to interact with unfolded target proteins. This may be the reason why αB-crystallin has an enhanced ability to prevent protein aggregation at higher temperature. Moreover, monomers as the active species may explain why αB-crystallin can prevent the aggregation of proteins at a very low stoichiometric ratio compared to the target protein [Hayashi J., Carver J.A. 2020]. It is suggested that this efficient stoichiometry is appropriate in the cell, as the destabilized target proteins are often present at low levels and aggregate slowly (for example within the eye lens).
The AC0 value equal to 91 is the maximum possible estimated value of chaperone capacity. DLS method is very sensitive to the aggregation process and formation of large particle, that is why we used it to study kinetics of target protein aggregation. However, the percent of very large particle in solution may be small (<1%). In the presence of high molecular mass aggregates, one could not see small oligomeric forms using DLS method. So, we use SV method to study small soluble associates/aggregates of proteins. Given that the aggregation process takes place over time, we select the maximum possible rotor speed. The large complexes HSPB5-aLa precipitated during the acceleration of the rotor, before the maximum rotor speed was reached. Their percentage was small (<5%) and they were not taken into account in the sedimentation velocity analysis.
Phb exists in the form of a dimer with a molecular mass of 195 kDa. aLa is a small protein with a molecular mass of 14.2 kDa. Molecular mass of HSPB5 monomer is 20 kDa. Therefore, the rotor speed in the case of aLa was higher than in the case of Phb, to detect the above mentioned small oligomeric forms and complexes with HSPB5.
The following text has been inserted on page 13, line 335:
However, the existence of high-order oligomeric HSPB5–αLa complexes cannot be ruled out. Such complexes may be present in DLS measurements. The typical particle size time-dependences for aggregates of αLa (1.0 mg/mL) in the absence or presence of HSPB5 are shown in Figure S2 of the Supplementary Materials.
Figure S2 (new) has been inserted in Supplementary Materials.
Comment 3
How were the samples processed for DLS analysis? Where the samples centrifuged or filtered before analysis?
The following text was added in 4.1 paragraph, line 580:
Stock Phb and HSPB5 solutions were centrifuged at 12,850 g and +4 °C for 10 min after the dialysis against 30 mM Hepes buffer, pH 6.8, containing 0.1 M NaCl, 0.2 mM EDTA.
The following text was added in 4.2 paragraph, line 594:
All solutions were prepared on the buffer passed through a 20 nm “Anotop” filter (Whatman).
Comment 4
Where the precipitated samples excluded from SV analysis?
Give the estimated molar mass values of the complexes between HSPB5 and Phb (Fig 9).
Reply
During the acceleration of the rotor, about 17% of the protein aggregated and precipitated before the maximum rotor speed was reached. These precipitated samples were excluded from SV analysis.
To estimate the molecular mass of Phb and possible complexes of Phb with HSPB5, the Svedberg equation was used:
M = sRT/D(1 - νρ),
(6)
where ν is the partial specific volume of a protein, ρ is solution density, R is molar gas constant, T is temperature in Kelvin, s is a sedimentation coefficient, D is a diffusion coefficient. For calculations we used sedimentation coefficients, determined by AUC, and diffusion coefficients (D), determined by DLS at 48 ºC.
The following text has been inserted in Section 2.5, line 352
The molecular mass estimate for the main peaks of Phb in the absence and presence of HSPB5 at concentrations of 0.025 and 0.075 mg/mL (Fig. 9) using Svedberg equation (see Section 4.3) gives the values: 205 ± 10, 234 ± 10 and 230 ± 10 kDa, respectively. For calculations, we used sedimentation coefficients determined by AUC and diffusion coefficients (D), determined by DLS at 48 ºC. The coefficients D for Phb and possible complexes of Phb with HSPB5 were estimated to be (6.5 ± 0.3)x10-7, (5.9 ± 0.5)x10-7, (6.2 ± 0.3)x10-7 cm2/s. The Rh values obtained by DLS method were equal to 6.2, 6.9 and 6.4 nm, respectively. The data of SV analysis and data obtained by DLS method ​​indicate the possibility of the formation of complexes between the dimeric form of the Phb and the monomeric or dimeric form of HSPB5.
The following text has been inserted at the end of Section 4.3. Analytical Ultracentrifugation (AUC)
To estimate the molecular mass of Phb and possible complexes of Phb with HSPB5, the Svedberg equation was used:
M = sRT/D(1 - νρ), (6)
where ν is the partial specific volume of a protein, ρ is solution density, R is molar gas constant, T is temperature in Kelvin, s is a sedimentation coefficient, D is a diffusion coefficient. For calculations we used sedimentation coefficients (s), determined by AUC, and diffusion coefficients (D), determined by DLS at 48 ºC.
Comment 5
Explain why the s20,W values for Phb with HSPB5 increase (Figure 9), while the Rh values show an apparent reduction in the size of Phb in the presence of HSPB5 (Figure 6).
Reply
The sedimentation coefficient value of main peak increases in the presence of HSPB5 (Figure 9), that might be explained by small increase in the molecular mass of the complex, on the one hand, and a change in the conformation of the target protein in the presence chaperone, on the other. The more compact conformation gives larger sedimentation coefficient for the same molecular mass.
Fig. 6 shows that the size of the start aggregates, with which the increase in the intensity (I - I0) of Phb aggregation begins, decreases in the presence of a chaperone. This may be explained by a decrease in the radius of the start aggregates (Rh,0) of Phb complexes with suboligomeric forms of chaperone compared to the start aggregates of the Phb alone.
Comment 6
- a) Can the fractions of SV analysis collected and run on SDS-PAGE gel or analyzed by Western blot analysis to validate the claims made in the paper?
- b) It would be helpful if Rh and Molar mass of the SV peaks is determined.
Reply
- a) Unfortunately, this is not possible.
- b) Rh and molecular mass of the SV peaks (Figure 9) have been determined (see response to comment 4). The following text was added in Section 2.5, lines 352-360.
Comment 7
The crowding effect on the HSPB5 chaperone function should also be tested with the Lysozyme substrate to see if the observed effect is substrate-dependent.
Reply
This is a very good idea. We plan such studies, but they will be the subject of our next article.
Comment 8
Words are merged in many places, especially in the introduction section.
Line 35 change “apaptosis” to “apoptosis.”
Reply
Corresponding correction has been made.
Line 35. “apaptosis” was changed to “apoptosis”.
Sincerely yours,
on behalf of the authors,
N.A. Chebotareva
Manuscript ID: ijms-843396
Title: Chaperone-like activity of HSPB5: the effects of quaternary structure dynamics and crowding
Natalia A. Chebotareva*, Svetlana G. Roman, Vera A. Borzova, Tatiana B. Eronina, Valeriya V. Mikhaylova and Boris I. Kurganov
Dear Ms. Jade Yu
Thank you very much for your letter of 2020-25-06 concerning our paper “Chaperone-like activity of HSPB5: the effects of quaternary structure dynamics and crowding”. We are thankful to you and the reviewers for the valuable comments. We have taken into account all the comments.
We send you the revised manuscript.
Reviewer #2
Dear Reviewer,
Thank you very much for your revision concerning our paper “Chaperone-like activity of HSPB5: the effects of quaternary structure dynamics and crowding”. We are thankful to you for the valuable comments took them into account by correcting our article.
Reviewers' comments:
This is a well-written manuscript describing clear results and making proper conclusions based on these results.
Specifically, the authors adequately establish a context for the issues addressed in the manuscript and these issues are important in our understanding of chaperone functions. The manuscript is data driven and has a clearly stated purpose.
The methods are adequate to answer the questions and contain the appropriate controls to produce proper conclusions.
Comment 1
Suggestion: My only comment on the methods would be to include specific experimental details in the descriptions.
Reply
The following text was added in 4.1 paragraph, line 566:
Stock Phb and HSPB5 solutions were centrifuged at 12,850 g and +4 °C for 10 min after the dialysis against 30 mM Hepes buffer, pH 6.8, containing 0.1 M NaCl, 0.2 mM EDTA.
The following text was added in 4.2 paragraph, line 579:
All solutions were prepared on the buffer passed through a 20 nm “Anotop” filter (Whatman).
The following text has been inserted at the end of Section 4.3. Analytical Ultracentrifugation (AUC):
To estimate the molecular mass of Phb and possible complexes of Phb with HSPB5, the Svedberg equation was used:
M = sRT/D(1 - νρ), (6) )
where ν is the partial specific volume of a protein, ρ is solution density, R is molar gas constant, T is temperature in Kelvin, s is a sedimentation coefficient, D is a diffusion coefficient. For calculations we used sedimentation coefficients (s), determined by AUC, and diffusion coefficients (D), determined by DLS at 48 °C.
Comment 2
The data are clearly presented and properly analyzed with appropriate statistical inferences when needed.
Suggestion: Although the recombinant proteins used in the study have been described in previous papers, I would suggest that the authors show coomassie stained gels of some of the protein fractions used for clarity and purity determination.
Reply
The following text has been included in Section 4.1, line 580:
Size-exclusion chromatography elution profile of HSPB5 on Superdex 200 column and SDS-gel electrophoresis of the stock HSPB5 sample and of its SEC elution fractions are presented in Figure S4. The authors thank Nikolai Sluchanko for a gift of the HSPB5 preparation and its characteristics.
Figure S4 was inserted into Supplementary Materials.
Figure S4. The characteristic of HSPB5 preparation. (A) The elution profile of HSPB5 obtained by SEC. (B) The SDS-gel electrophoresis of the stock HSPB5 preparation (second lane from the left) and the fractions from peak in panel A in the elution times interval 48-61 min. The first lane from the left are MW standards.
The authors thank Nikolai Sluchanko for a gift of the HSPB5 preparation and its characteristics.
SDS-gel electrophoresis of Phb sample used in the experiments.
Comment 3
The conclusions follow the data presented and are reasonable interpretations of the experimental results.
Suggestion: The discussion can be revised to be more concise and less descriptive.
Reply
The discussion has been revised.
Line 443, the sentence: The polydispersity of some sHSPs, such as HSPB5 (aB-crystallin), depends on the exchange of subunits, which occurs through (partial) dissociation of the oligomer, - has been deleted
Line 453, part of the sentence “By the tandem mass spectrometry (MS/MS) method” – was removed
Line 475, the sentence “In this case, a shift occurs in the equilibrium between oligomers and (monomeric + dimeric) substructures” -was removed.
Comment 4
Overall the manuscript describes the effects of crowding on the anti-aggregation activity of a sHSP and the results provide a better understanding of the processed involved.
Suggestion: However, it would be advisable to alter the language of the major conclusion to make it less definite.
Reply
We tried to do it, but it is difficult.
Sincerely yours,
on behalf of the authors,
N.A. Chebotareva

Reviewer 2 Report
This is a well-written manuscript describing clear results and making proper conclusions based on these results.
Specifically, the authors adequately establish a context for the issues addressed in the manuscript and these issues are important in our understanding of chaperone functions. The manuscript is data driven and has a clearly stated purpose.
The methods are adequate to answer the questions and contain the appropriate controls to produce proper conclusions.
Suggestion: My only comment on the methods would be to include specific experimental details in the descriptions.
The data are clearly presented and properly analyzed with appropriate statistical inferences when needed.
Suggestion: Although the recombinant proteins used in the study have been described in previous papers, I would suggest that the authors show coomassie stained gels of some of the protein fractions used for clarity and purity determination.
The conclusions follow the data presented and are reasonable interpretations of the experimental results.
Suggestion: The discussion can be revised to be more concise and less descriptive.
Overall the manuscript describes the effects of crowding on the anti-aggregation activity of a sHSP and the results provide a better understanding of the processed involved.
Suggestion: However, it would be advisable to alter the language of the major conclusion to make it less definite.
Author Response
N.A. Chebotareva
Manuscript ID: ijms-843396
Title: Chaperone-like activity of HSPB5: the effects of quaternary structure dynamics and crowding
Natalia A. Chebotareva*, Svetlana G. Roman, Vera A. Borzova, Tatiana B. Eronina, Valeriya V. Mikhaylova and Boris I. Kurganov
Dear Ms. Jade Yu
Thank you very much for your letter of 2020-25-06 concerning our paper “Chaperone-like activity of HSPB5: the effects of quaternary structure dynamics and crowding”. We are thankful to you and the reviewers for the valuable comments. We have taken into account all the comments.
We send you the revised manuscript.
Reviewer #2
Dear Reviewer,
Thank you very much for your revision concerning our paper “Chaperone-like activity of HSPB5: the effects of quaternary structure dynamics and crowding”. We are thankful to you for the valuable comments took them into account by correcting our article.
Reviewers' comments:
This is a well-written manuscript describing clear results and making proper conclusions based on these results.
Specifically, the authors adequately establish a context for the issues addressed in the manuscript and these issues are important in our understanding of chaperone functions. The manuscript is data driven and has a clearly stated purpose.
The methods are adequate to answer the questions and contain the appropriate controls to produce proper conclusions.
Comment 1
Suggestion: My only comment on the methods would be to include specific experimental details in the descriptions.
Reply
The following text was added in 4.1 paragraph, line 566:
Stock Phb and HSPB5 solutions were centrifuged at 12,850 g and +4 °C for 10 min after the dialysis against 30 mM Hepes buffer, pH 6.8, containing 0.1 M NaCl, 0.2 mM EDTA.
The following text was added in 4.2 paragraph, line 579:
All solutions were prepared on the buffer passed through a 20 nm “Anotop” filter (Whatman).
The following text has been inserted at the end of Section 4.3. Analytical Ultracentrifugation (AUC):
To estimate the molecular mass of Phb and possible complexes of Phb with HSPB5, the Svedberg equation was used:
M = sRT/D(1 - νρ), (6) )
where ν is the partial specific volume of a protein, ρ is solution density, R is molar gas constant, T is temperature in Kelvin, s is a sedimentation coefficient, D is a diffusion coefficient. For calculations we used sedimentation coefficients (s), determined by AUC, and diffusion coefficients (D), determined by DLS at 48 °C.
Comment 2
The data are clearly presented and properly analyzed with appropriate statistical inferences when needed.
Suggestion: Although the recombinant proteins used in the study have been described in previous papers, I would suggest that the authors show coomassie stained gels of some of the protein fractions used for clarity and purity determination.
Reply
The following text has been included in Section 4.1, line 580:
Size-exclusion chromatography elution profile of HSPB5 on Superdex 200 column and SDS-gel electrophoresis of the stock HSPB5 sample and of its SEC elution fractions are presented in Figure S4. The authors thank Nikolai Sluchanko for a gift of the HSPB5 preparation and its characteristics.
Figure S4 was inserted into Supplementary Materials.
Figure S4. The characteristic of HSPB5 preparation. (A) The elution profile of HSPB5 obtained by SEC. (B) The SDS-gel electrophoresis of the stock HSPB5 preparation (second lane from the left) and the fractions from peak in panel A in the elution times interval 48-61 min. The first lane from the left are MW standards.
The authors thank Nikolai Sluchanko for a gift of the HSPB5 preparation and its characteristics.
SDS-gel electrophoresis of Phb sample used in the experiments.
Comment 3
The conclusions follow the data presented and are reasonable interpretations of the experimental results.
Suggestion: The discussion can be revised to be more concise and less descriptive.
Reply
The discussion has been revised.
Line 443, the sentence: The polydispersity of some sHSPs, such as HSPB5 (aB-crystallin), depends on the exchange of subunits, which occurs through (partial) dissociation of the oligomer, - has been deleted
Line 453, part of the sentence “By the tandem mass spectrometry (MS/MS) method” – was removed
Line 475, the sentence “In this case, a shift occurs in the equilibrium between oligomers and (monomeric + dimeric) substructures” -was removed.
Comment 4
Overall the manuscript describes the effects of crowding on the anti-aggregation activity of a sHSP and the results provide a better understanding of the processed involved.
Suggestion: However, it would be advisable to alter the language of the major conclusion to make it less definite.
Reply
We tried to do it, but it is difficult.
Sincerely yours,
on behalf of the authors,
N.A. Chebotareva
Please see the attachment

Round 2
Reviewer 1 Report
The revised manuscript has improved significantly. The authors have responded to all queries satisfactorily.